Sea urchin harvest inside marine protected areas: an opportunity to investigate the effects of exploitation where trophic upgrading is achieved

Ceccherelli Giulia cecche@uniss.it 1
Addis Piero 2
Atzori Fabrizio 3
Cadoni Nicoletta 3
Casu Marco 4
Coppa Stefania 5
De Luca Mario 1
de Lucia Giuseppe Andrea 5
Farina Simone 6 7
Fois Nicola 8
Frau Francesca 3
Gazale Vittorio 9
Grech Daniele 6
Guala Ivan 6
Mariani Mariano 10
Marras Massimo SG 11
Navone Augusto 12
Pansini Arianna 1
Panzalis Pieraugusto 12
Pinna Federico 1
Ruiu Alberto 10
Scarpa Fabio 4
Piazzi Luigi 1
1 Dipartimento di Chimica e Farmacia, Universitá di Sassari, Via Piandanna , Sassari , Italy
2 Dipartimento di Scienze della Vita e dell’Ambiente, Universitá di Cagliari, Via Fiorelli , Cagliari , Italy
3 Capo Carbonara –Villasimius Marine Protected Area, Via Roma , Villasimius (CA) , Italy
4 Dipartimento di Medicina Veterinaria –Sez. Fisiologia della Nutrizione e Zoologia, Universitá di Sassari , Sassari , Italy
5 Istituto per lo studio degli Impatti Antropici e Sostenibilità in ambiente marino (IAS) –Consiglio Nazionale delle Ricerche (CNR), Loc. Sa Mardini , Torre Grande (OR) , Italy
6 IMC –International Marine Centre, Loc. Sa Mardini , Torre Grande, OR , Italy
7 SZN –Stazione Zoologica Anton Dohrn , Villa Comunale Napoli , Italy
8 Agris –Agricultural Research Agency of Sardinia –Bonassai SS , Sassari , Italy
9 Isola dell’Asinara Marine Protected Area, via Ponte Romano , Porto Torres (SS) , Italy
10 Capo Caccia –Isola Piana Marine Protected Area, Loc. Tramariglio SP , Alghero, SS , Italy
11 Penisola del Sinis –Isola di Mal di Ventre Marine Protected Area, Corso Italia , Cabras, OR , Italy
12 Tavolara Punta Coda Cavallo Marine Protected Area, Via S. Giovanni , Olbia (SS) , Italy
Gandini Patricia
Electronic publication date: 2022 Mar 7
Publication date: 2022
Volume: 10
Electronic Location ID: e12971
Received 2021 Jul 15; Accepted 2022 Jan 30
Copyright: ©2022 Ceccherelli et al.
Copyright year: 2022
Copyright holder: Ceccherelli et al.
License: This is an open access article distributed under the terms of the Creative Commons Attribution License, which permits unrestricted use, distribution, reproduction and adaptation in any medium and for any purpose provided that it is properly attributed. For attribution, the original author(s), title, publication source (PeerJ) and either DOI or URL of the article must be cited.
License URL: https://creativecommons.org/licenses/by/4.0/

Keywords: Cumulative effects, Exploitation, Coastal management, Marine protected areas, Mediterranean Sea, Predation, Harvest, Restrictions, Multiple use areas, Natural predators

Funding: AGRIS Sardegna The Sardinian Regional Government Italian Ministry of the Environment The Interreg V a Italy France Maritime 2014-2020 Cooperation Program This research was funded by AGRIS Sardegna, supported by the Sardinian Regional Government, and by various commitments of the Sardinian MPAs funded by the Italian Ministry of the Environment. IG and SF were funded by the Interreg V a Italy France Maritime 2014-2020 Cooperation Program, project “Gestione Integrata delle Reti ecologiche attraverso i Parchi e le Aree Marine - GIREPAM” (Asse 2 - Lotto 3 - PI 6C-OS 1). The funders had no role in study design, data collection and analysis, decision to publish, or preparation of the manuscript.

==============================
Background

Marine protected areas (MPAs) usually have both positive effects of protection for the fisheries’ target species and indirect negative effects for sea urchins. Moreover, often in MPAs sea urchin human harvest is restricted, but allowed. This study is aimed at estimating the effect of human harvest of the sea urchin Paracentrotus lividus within MPAs, where fish exploitation is restricted and its density is already controlled by a higher natural predation risk. The prediction we formulated was that the lowest densities of commercial sea urchins would be found where human harvest is allowed and where the harvest is restricted, compared to where the harvest is forbidden.

Methods

At this aim, a collaborative database gained across five MPAs in Sardinia (Western Mediterranean, Italy) and areas outside was gathered collecting sea urchin abundance and size data in a total of 106 sites at different degrees of sea urchin exploitation: no, restricted and unrestricted harvest sites (NH, RH and UH, respectively). Furthermore, as estimates made in past monitoring efforts (since 2005) were available for 75 of the sampled sites, for each of the different levels of exploitation, the rate of variation in the total sea urchin density was also estimated.

Results

Results have highlighted that the lowest sea urchin total and commercial density was found in RH sites, likely for the cumulative effects of human harvest and natural predation. The overall rate of change in sea urchin density over time indicates that only NH conditions promoted the increase of sea urchin abundance and that current local management of the MPAs has driven towards an important regression of populations, by allowing the harvest. Overall, results suggest that complex mechanisms, including synergistic effects between natural biotic interactions and human pressures, may occur on sea urchin populations and the assessment of MPA effects on P. lividus populations would be crucial to guide management decisions on regulating harvest permits. Overall, the need to ban sea urchin harvest in the MPAs to avoid extreme reductions is encouraged, as inside the MPAs sea urchin populations are likely under natural predation pressures for the trophic upgrading.

Introduction

Unsustainable harvesting is now one of the prevalent issues affecting threatened marine species (Di Minin et al., 2019). In fact, about one-third of commercial wild fish stocks are currently being overfished (FAO, 2016) and global marine fishery catches are declining (Pauly & Zeller, 2016). The general overexploitation of species has consequences on food webs and ecosystem functioning, firstly due largely to the widespread declines in marine predators (Strong & Frank, 2010). When high trophic level predators are removed from ecosystems the ‘trophic downgrading’ (sensu Estes et al., 2011) leaves greater proportions of low trophic level species. This leads to an imbalance in the regulation of coastal systems (Britten et al., 2014).

Marine protected areas (MPAs) have emerged as a promising management tool for the conservation and recovery of marine coastal ecosystems (Russ et al., 2004; Gaines et al., 2010; Giakoumi et al., 2017). Although it was evidenced that the enforcement level coupled with an effective surveillance is the essential requirement to reestablish the predatory interactions that have been lost (Guidetti, 2006; Giakoumi et al., 2017; Giakoumi et al., 2018), encouraging data for managing the marine coastal environments have been provided even by small, well-enforced, fully protected (integral reserve) areas that have had relevant ecological effects (i.e., Guidetti et al., 2008; Giakoumi et al., 2017). In effective MPAs, the density of prey species, whether fish or invertebrates, can be drastically reduced by the increased abundance of predators (e.g., Willis & Anderson, 2003; Sala et al., 2013; Giakoumi et al., 2017; Guidetti et al., 2019).

In temperate reefs, sea urchins have been identified as important players in trophic cascades (Pinnegar et al., 2000; Filbee-Dexter & Scheibling, 2014; Ling et al., 2015; Carr & Reed, 2016; Melis et al., 2019). Despite the importance of recruitment and environmental conditions such as refuge availability, storms and temperature (Hereu et al., 2012; Clemente et al., 2013; Yeruham et al., 2015; Oliva et al., 2016; Piazzi & Ceccherelli, 2017; Medrano et al., 2019; Farina et al., 2020), sea urchin density is mainly controlled by predation through consumptive and non-consumptive effects (e.g., Sih, Englund & Wooster, 1998; Guidetti, 2006; McClananhan, Verheij & Maina, 2006; Hernández et al., 2007; Guidetti & Sala, 2007; Seytre et al., 2013; Pessarrodona et al., 2019). On the other hand, the benthic community structure is controlled by sea urchin grazing effects: models demonstrated the importance of feedback loops that stabilize each state of community structure and that transitions between these states have different thresholds when moving from kelp forest to urchin barrens or barrens to kelp forest (e.g., Filbee-Dexter & Scheibling, 2014). Therefore, depending on the system, sea urchin density is managed to be either reduced to relieve macrophyte forests from grazing (e.g., Ling, Ibbott & Sanderson, 2010; Piazzi & Ceccherelli, 2019) or restored to prevent the excessive proliferation of macroalgae (such as in coral reefs, e.g., Nozawa, Lin & Meng, 2020; Dang et al., 2020; Williams, 2021). On the other hand, in some systems sea urchin restocking projects are wanted to simply reverse the depletion of wild stocks and prevent the collapse of the sea urchin fishery due to overexploitation, regardless the effects on the benthic biodiversity (Couvray et al., 2015; De la Uz et al., 2018; Giglioli et al., 2021).

In the Mediterranean, the sea urchin Paracentrotus lividus (Lamarck, 1816) (Echinoidea: Parechinidae) is a key herbivore of the shallow subtidal rocky habitats, playing a central role in the trophic cascade (Micheli et al., 2005; Giakoumi et al., 2012). Its high-density populations, due to the lack of predators (overexploited areas), can have dramatic effects on rocky macroalgal community biodiversity, producing barren grounds (Guidetti & Dulcic, 2007; Gianguzza et al., 2011; Boada et al., 2017). However, in MPAs where the recovery of natural predators is widely achieved (Pinnegar et al., 2000; Guidetti et al., 2014), sea urchin abundance can be shaped by multiple predator effects, due to fish (Guidetti, Bussotti & Boero, 2005) and benthic invertebrates (Bonaviri et al., 2009; Bonaviri et al., 2012; Boada et al., 2015; Farina et al., 2016). Predation mostly affects sea urchins up to 40 mm in size (Pessarrodona et al., 2019), therefore reserve effectiveness usually leads to higher frequency of large P. lividus compared to no protected areas (Hereu et al., 2005; Loi et al., 2017).

However, natural predation is not the only pressure controlling P. lividus populations. Since it is an edible species, human harvest of this sea urchin has recently intensified (Furesi et al., 2016), making P. lividus one of the most exploited benthic invertebrate species in the Mediterranean (Gianguzza et al., 2006; Ceccherelli et al., 2011). Thus, there is growing interest concerning the maintenance of P. lividus populations both for ecological and commercial aims. This interest has led to an increase in the knowledge of mechanisms regulating P. lividus abundance (e.g., Guidetti, 2004; Hereu, Zabala & Sala, 2008; Ceccherelli, Pinna & Sechi, 2009; Prado et al., 2012; Boada et al., 2015; Oliva et al., 2016; Farina et al., 2018) in order to define sustainable harvest (e.g. Ceccherelli et al., 2011; Bertocci et al., 2014). The current regulation varies on a local scale and acts on the harvest season, the number of fishermen, the minimum size. Thus, the perception of an inherent trade-off between achieving conservation and fishing goals may be far from being reduced or eliminated (Gaines et al., 2010). Particularly severe conflicting aims may arise from multiple-use MPAs: their goal is to allow a variety of human activities that are managed comprehensively to support compatible uses while at the same time protecting key habitats and resources. In these areas, management decisions need to accurately consider the potential conflict between human demands and biodiversity conservation. Therefore, data providing important feedbacks on the potential interaction between conservation goals and human activities are needed to either review the decisions or readdress the objectives of the MPA, according to the principles of adaptive management (Agardy et al., 2003; Pomeroy et al., 2005).

In Sardinia (central-western Mediterranean Sea, Italy), sea urchin roe is a common ingredient in several dishes and harvesting has been historically carried out both by professional and non-professional fishermen (Furesi et al., 2016). To minimise the risk of overexploitation the regional government of Sardinia has imposed restrictions on the commercial fishing of the edible sea urchin: (i) by granting a limited number of firms (about 200) authorisation for this activity; (ii) by limiting the fishing season from November to May; (iii) by fixing daily catch quotas (1,500 to 3,000 sea urchins per day for each professional fisherman); and (iv) by providing a minimum sea urchin size to harvest (test diameter of 50 mm). These restrictions encompass all Sardinia coasts with the inclusion of the MPAs, where a specific number of authorized fishermen have been addressed (Table 1). However, each MPA, based on its own management plan, has independently allowed harvesting in partially protected zones (both B and C zones, according to the Italian designation) by allowing a fixed number of authorized professional fishermen (Fig. 1), who are not given access to the fully protected zone (integral reserves, A zone) and harvest has never been allowed (Table 1). Therefore, the effectiveness of protection is not obvious since the effects of human and natural predation may detrimentally cumulate inside the MPAs where sea urchin harvest is allowed. Moreover, the pressures, natural predation and human harvest, should have the different shaping effect on the sea urchin population structure, as humans should exploit only large-sized individuals because of regulations (>50 mm) while fish attacks are more frequent on small-sized urchins (Guidetti, 2004). Thus, complex mechanisms may occur and the assessment of MPA effects on P. lividus populations is crucial to guide management decisions on regulating sea urchin harvest permits (Coppa et al., 2021), with the intention of avoiding their local strong reduction. In particular, in Sardinia the demand for P. lividus has grown significantly in the last two decades (Furesi et al., 2016) so that studies have focused on the effect of harvest restrictions (i.e., Pais et al., 2007; Ceccherelli et al., 2011; Loi et al., 2017), and data are now available for observing patterns of change depending on the management. In this way, Sardinia may represent a suitable case to assess the management effectiveness of MPA in the maintenance of sea urchin populations in order to obtaining useful information for the conservation of ecosystems functioning and the sustainable harvest of the resource.

Table 1 MPA features in terms of: year of establishment and relative zonation (i.e. surface covered by different degree of protection); regulation of sea urchin harvest and fishery.

	SN MPA	CI MPA	AS MPA	TV MPA	CC MPA	
Year of establishment	1997	2002	1997	1997	1998	
Area protected in A zone (Km2)	3.5	0.4	5.4	5.3	1.0	
Area protected in B zone (Km2)	9.7	4.1	70.2	25.6	16.9	
Area protected in C zone (Km2)	229.2	20.5	32.4	127.9	66.0	
Urchin harvest in A zone	NH	NH	NH	NH	NH	
Urchin harvest in B zone	NH	RH	NH	RH	NH	
Urchin harvest in C zone	RH	RH	NH	RH	NH	
# Authorized urchin fishers
through years in RH	282 (2005)
276 (2007)
184 (2012)
119(2015)
74 (2018)	–
10 (2007)
10 (2012)
10 (2015)
10 (2018)	None
None
None
None
None	–
–
–
17 (2015)
14 (2018)	None
None
None
None
None	
	40 (2019)	10 (2019)	None	2 (2019)	None	
Fishery of natural predators in A zone	Not allowed	Not allowed	Not allowed	Not allowed	Not allowed	
Fishery of natural predators in B zone	Allowed	Allowed	Allowed	Allowed	Allowed	
Fishery of natural predators in C zone	Allowed	Allowed	Allowed	Allowed	Allowed	
# NH sites for the spatial evaluation	6	1	19	2	5	
# RH sites for the spatial evaluation	8	9	–	20	–	
Notes.

SN, Penisola del Sinis-Isola di Mal di Ventre; CI, Capo Caccia-Isola Piana; AS, Isola dellAsinara; TV, Tavolara Punta Coda Cavallo and CC, Capo Carbonara. For each MPA the # of NH (no harvest = not allowed) and RH (restricted harvest = allowed, but restricted) sites useful for the current evaluation are also given.

Figure 1 Maps of Sardinia and the studied MPAs. CI, Capo Caccia-Isola Piana; AS, Isola dell’Asinara; SN, Penisola del Sinis-Isola di Mal di Ventre; TV, Tavolara Punta Coda Cavallo; CC, Capo Carbonara.

Zonation of each MPA is showed: A zone in red, B zone in grey and C zone in light grey.

The present study was designed to examine the effect of human harvest of an herbivore sea urchin in no take areas within MPAs, where fish exploitation is restricted and thus its density is already controlled by a higher natural predation risk. At this aim we compared the P. lividus total (any size urchin) and commercial (only urchins larger than 50 mm) density in different harvest conditions in Sardinia (Italy). This goal was achieved by two approaches: a spatial evaluation and a temporal evaluation. For the spatial evaluation, we compared total and commercial P. lividus density at sites with different harvest regulations using data collected in 2018–2019. The prediction we formulated was that the lowest densities of commercial sea urchins would be found where human harvest is allowed and where the harvest is restricted, compared to where the harvest is forbidden. Our prediction for the whole population of P. lividus was that the lowest total density of sea urchins would be found in protected zones where the effect of human harvest (even if restricted) and natural predation can cumulate. For the temporal evaluation, only the sites where past monitoring data on P. lividus density were available (overall since 2005) were considered, so that the variability of the total density of P. lividus over time was also estimated depending on the harvest level, with the expectation that the greatest changes would be found in areas where both human and natural predation have occurred. For both evaluations, a collaborative database was produced by integrating information taken at different times by several research institutions involved in monitoring Sardinian MPAs and non-protected areas. Results may inform about drivers of overexploitation of sea urchins and the management of multiple-use MPAs. Implications of the study regard the addressing both by the MPAs and Regional management of the sea urchin harvest and may also give insights into the successful management of any sustainable exploitation of resource in danger of being depleted.

Materials & Methods

Study locations

P. lividus density and size were assessed at 106 sites located along the coast of Sardinia (Italy, Mediterranean Sea, Fig. 1, Fig. S1–S5), each site corresponding to about 200 m of coastline: 33 no harvest sites (NH, harvest never been allowed), 37 restricted harvest sites (RH, exploitation of sea urchins is restricted by the MPA) and 36 harvested unrestricted sites (UH, no limits for an overall exploitation, outside the MPA) sites. The sites considered are located in all geographical areas within the Island for all NH, RH and UH (Figs. S1–S5), with a wide natural range of environmental conditions (e.g., wave exposure, slope, mineralogy and complexity of the bottom, etc.). NH and RH sites from the main Sardinian MPAs were included (Table 1): Penisola del Sinis-Isola di Mal di Ventre (SN MPA), Capo Caccia-Isola Piana (CI MPA), Isola dell’Asinara (AS MPA), Tavolara Punta Coda Cavallo (TV MPA), and Capo Carbonara (CC MPA). These multiple-use MPAs differ in their establishment dates, in the extension of zones with different degrees of protection (A, B and C zones), in reserve effectiveness and in their management of the sea urchin harvest (Table 1). However, their no-take areas (A zones) have recently shown a higher biomass of commercial fish compared to other protected areas (Natura 2000 sites) or unprotected sites (Guidetti et al., 2019). In particular, in the MPAs a higher biomass of the urchin predators, the fish Diplodus sargus and Diplodus vulgaris, was found compared to outside the MPAs, about 1,600 g/125 m2 and 500 g/125 m2 in mean, respectively. Based on these estimates we have formulated the hypothesis of a lower natural predation at the UH sites, rather than inside the MPAs (NH and RH sites).

Each MPA management body has authorized a number of fishermen to harvest sea urchins inside the boundaries, therefore affecting pressure on the local P. lividus density and territorial use rights, as for example in the SN MPA (Coppa et al., 2021) and TV MPA (Table 1). Furthermore, in MPAs where sea urchin harvest is allowed to fishermen the number of catches is monitored, but the data collected in the logbooks are unfortunately not reliable (e.g., Coppa et al., 2021). UH sites are located outside the MPAs, where the exploitation of sea urchins is regulated by the regional government, and urchins can be harvested by both professional and recreational fishermen (i.e., any permanent resident in Sardinia, is allowed to harvest up to 50 sea urchins per day). Therefore, outside the MPAs, sea urchins can indeed be harvested without actual limitations, as restrictions set about the sea urchin size, catch quotas and harvest season have nothing to do with the overall number of urchins effectively removed from a site which is often determined only by the accessibility of sites (Ceccherelli et al., 2011).

Data collection

For the spatial evaluation, sampling was done between May 2018 and November 2019 at 106 sites (33, 37, and 36 for NH, RH and UH, respectively). At each site, sea urchin abundance was estimated in the field by scuba divers counting all individual within 1 m2 frames (haphazardly placed meters of distance apart) and the size of each individual (test diameter) in the quadrat was measured with callipers to the closest mm. In each quadrat all sea urchins larger than 20 mm in test diameter were quantified in all the crevices and under the boulders, even turning over the stones within the quadrat. The complexity of the bottom basically changed according to the site mineralogy (granite, sandstone, basaltic and limestone) which depended on the geographical sector of the Island: because either NH, RH, and UH sites were considered from different geographical areas (Figs. S1–S5), we assumed that a natural range in bottom complexity was considered for each harvest type. To obtain comparable data, values collected on rocky substratum at a depth of 5 m were selected from the available dataset: this led to dealing with a different number of quadrats (from 10 to 30) for each site, for an overall dataset of 886 quadrats.

For the temporal evaluation, only the sites for which past monitoring data (since 2005) were available were considered. Among these, based on the consistency in the sampling methods used (5 m of depth using 1 m2 quadrat size), 75 were ultimately selected (32, 28, and 15 for NH, RH and UH, respectively).

For both evaluations the short temporal variability due to the month of sampling was neglected and data analyses were done on the average values obtained from quadrats (the sites were replicates). This allowed us to limit the exclusion of data (due to the unbalanced number of replicate quadrats), though it has prevented us from estimating the variability at the scale of the site.

Data analyses

Data were analyzed using univariate permutational analyses of variance based on Euclidean distance measure (Terlizzi et al., 2007; Anderson, 2001). In this analysis, P-values associated with F statistics are obtained by permutation (Clarke & Gorley, 2006). For the spatial evaluation, two one-way univariate permutational analyses of variance (PERMANOVAs) were run to estimate the effects of the factor ‘harvesting conditions’ (NH, RH, and UH) on the total and commercial sea urchin densities where the mean value at each site was used as replicate (n = 33, 37, and 36, respectively). A posteriori, pair-wise tests were run to identify alternative hypotheses. For the temporal evaluation, a one-way univariate PERMANOVA was run to compare the rate of change in total urchin density (calculated across years) at the different harvesting conditions (NH, RH, and UH), where the mean change/year at each site was used as replicate (32, 28, and 15, respectively).

The effect of the harvesting conditions was also visually examined by quantifying the natural logarithm of the ratio between the values of each response variable (i.e., sea urchin total and commercial density) at NH and RH conditions versus UH conditions (response ratio ln RR, Micheli et al., 2004). With this approach, the observed effect is independent of the absolute density at each location. Positive RRs indicate greater values under either NH or RH than in UH conditions, whereas negative values indicate greater values in UH than in protected conditions (NH or RH). A ratio of zero, instead, means that similar values were found between protected and control conditions.

Results

The spatial evaluation has evidenced that P. lividus total and commercial densities were significantly affected by the harvesting conditions, where NH is the condition that promoted the highest response (Fig. 2 and Table 2). Particularly, the lowest total density of urchins was found inside the MPAs in the RH, rather than in the NH and UH, while commercial P. lividus was differently affected by the harvest: commercial density was ranked among harvesting conditions as being highest in the NH and lowest in the RH. In terms of ln RR, both sea urchin variables had a positive response in NH and a negative one in RH conditions (Fig. 3).

Figure 2 Paracentrotus lividus. (A) total and (B) commercial density (mean and the confidence interval) at the NH (no harvest), RH (restricted harvest) and UH (unrestricted harvest) sites (replicated sites n = 33, 37, and 36, respectively).

Table 2 Spatial evaluation: PERMANOVA results on the effect of harvest type on total P. lividus density and commercial density.

		Total
abundance	Commercial abundance	
	df	MS	Pseudo-F	MS	Pseudo-F	
Harvest	2	34.99	6.71	12.30	8.30	
Residual	103	5.21		1.48		
Pair wise tests		NH=UH>RH	NH>UH>RH	
Notes.

Significant results (p < 0.05) are given in bold. Harvest: NH, no harvest; RH, restricted harvest and UH, unrestricted harvest.

Figure 3 Paracentrotus lividus. Response (mean ± SE) of total density (black) and commercial density (white) to no harvest (NH) and restricted harvest (RH) conditions.

Response was quantified by the natural logarithm of the ratio between the values of each response variable (sea urchin total and commercial density) at NH and RH conditions versus UH conditions.

The temporal variability in the total urchin density has evidenced an overall decrease in sea urchin density since 2005 in NH, as well as in RH and UH conditions. This general pattern derives from a graphical inspection of total urchin density in each condition over time (obtained by averaging all data available from all sites sampled), although the sample size has changed considerably (Fig. 4). However, the comparison of the rate of temporal variability in total P. lividus density (change in ind/yr per site) among conditions, has revealed different effects of harvesting conditions although the high variability. In fact, there has been an overall increase (positive effect) in mean density per year in NH, whereas in RH and UH the urchin density has decreased (negative effect) in a similar manner (Table 3 and Fig. 5).

Figure 4 Paracentrotus lividus. Temporal variability in total abundance (individuals/m2) at NH, no harvest; RH, restricted harvest and UH, unrestricted harvest conditions from 2005 to 2019.

The number of sites from which each mean (and SE) was calculated is indicated at the bottom of the plot.

Table 3 Temporal evaluation: PERMANOVA results on the effect of harvest type on the variation rate of total P. lividus density (yr−1).

	Variation in total abundance	
	df	MS	Pseudo-F	P	
Harvest	2	1.51	4.54	0.014	
Residual	72	0.33			
Pair wise test	NH>RH=UH	
Notes.

Harvest: NH, no harvest; RH, restricted harvest and UH = unrestricted harvest.

Figure 5 Paracentrotus lividus. Rate of variation (mean±SE across 15 years) in total abundance (individuals/m2yr) at NH (no harvest), RH (restricted harvest) and UH (unrestricted harvest) conditions (replicated sites n = 27, 28, and 13, res.

Discussion

This collaborative effort highlighted differences in total and commercial sea urchin density among different harvesting conditions. The spatial evaluation showed that commercial P. lividus density was significantly higher in NH than in UH conditions, while no differences were detected between the two conditions for total density, suggesting that natural predation can enhance the abundance of large-sized individuals by affecting the smaller individuals (Pessarrodona et al., 2019). Moreover, values of both total and commercial sea urchin density were lower in the RH condition. Moreover, the RH effect through time on total and commercial P. lividus density was again of higher reduction compared to the other harvest types, indicating that RH is the condition where the density of urchins and the commercial portion have regressed the most. Moreover, the same indication was gained from ln RR, as negative values (for both variables) were only found for the RH condition. Overall these findings suggest that the local management of the MPAs (by authorizing sea urchin harvest and not predatory fish depletion) has driven the urchin density to be lower than in unprotected areas (UH). Therefore, the prediction that in RH conditions human harvest effects would potentially cumulate with natural predation effects seems to be met. This finding offers a further example of the unexpected effect of fishing when natural trophic webs are not considered (Scheffer, Carpenter & de Young, 2005). This phenomenon has been observed worldwide (Daskalov, 2002; Loh et al., 2015; Stuhldreier et al., 2015) and the increase in resource depletion has led to the development of holistic approaches to fisheries management (ecosystem-based fisheries management, (Link, 2010) and multispecies models of predation, where trophic pressures vary and the assumption of constant natural mortality is neglected (Jurado-Molina, Livingston & Ianelli, 2005; Kinzey & Punt, 2009). Moreover, recent investigations have explicitly highlighted how ignoring natural trophic interactions affects stock assessment model performance and fisheries management (Farina et al., 2020; Trijoulet, Fay & Miller, 2020).

Understanding the P. lividus spatial variability in Sardinia can be greatly facilitated by the analysis of changes in density over time. Although there were wide differences in the encompassed time, a general overall decreasing pattern was observed. However, the analysis has evidenced that the temporal variability depended on the harvesting conditions, with a decreased rate in RH and UH conditions and an increased rate only at NH. The positive NH effect give us two important indications: firstly, this finding suggests that totally banning the harvest (and probably enforcement) has allowed P. lividus density to increase; secondly it cannot be claimed that environmental threats, such as global warming or acidification (Yeruham et al., 2015; Asnaghi et al., 2013; Asnaghi et al., 2020), can justify the low sea urchins density found in RH in 2019, as a density reduction would have been found at all harvest conditions. Furthermore, sea urchin density (e.g. recruitment, predation risk) are enormously affected by context-dependent features such as nutrient supply (Boada et al., 2017), larvae dispersion (Oliva et al., 2016), and mostly substrate characteristics, which may influence recruitment and predation risk (Pinna et al., 2012; Prado et al., 2012). Thus, one could argue that the physical characteristics of the sites could have contributed to the observed patterns. However, because there was a large natural features variability among sites within each harvest condition (e.g. in terms of wave exposure and bottom types), in this study case harvest restriction was inferred to be the most relevant driver affecting sea urchin density.

Furthermore, the high decrease in urchin density occurred over time is worrisome for the species conservation, as the general depletion of urchins may lead to a stable state by triggering some feed-back mechanisms (Filbee-Dexter & Scheibling, 2014; Guarnieri et al., 2020). The density of P. lividus individuals of commercial size depends on the recruitment success and thus it is constrained by natural predation pressure (targeting the small-sized individuals, Guidetti & Sala, 2007). At the same time, recruitment depends on the occurrence of breeding individuals (Oliva et al., 2016) and becomes impeded by human harvest (which removes the most effective reproducers, Loi et al., 2017). This scenario highlights the need for more in-depth knowledge of the fate of P. lividus coastal populations. In fact, although harvest and natural predation affect different portions of the same sea urchin populations, they might also be seen as synergistic pressures on the same resource. Overall, the rate of change in sea urchin density in RH and UH conditions claim for a better management: for the former, by effectively excluding or reducing human harvest within the MPAs in order to restore the urchin populations, and for the latter, by defining harvest limits for a sustainable exploitation.

The use of MPAs for fisheries management has become popular in the last decades (e.g., Pelc et al., 2010; Di Lorenzo et al., 2020) as they can enhance fishery yield and improve stock sustainability through spillover effects. They increase adult density, and thus the production of eggs and larvae. However, to obtain such goals, an assessment of the spatial distribution of resources such as sea urchins, that have a complex spatial structure, is fundamental for addressing the spatial scales of management (Ouréns, Naya & Freire, 2015). Therefore, fishery management failures are often due to a mismatch between the spatial scale of exploited populations and the scale of their management. This might be the case in Sardinia, where wide areas of most of the MPAs (TV, CI and SN) have been offered to the fisheries for several years regardless of the size of the stocks actually available and the identification of metapopulations.

As a matter of fact, many wildlife populations are now well below equilibrium levels in many industrial countries, and humans provide their predominant preying control as food webs are so depleted that natural predators are lacking (Strong & Frank, 2010). However, humans can regulate themselves by defining catch limits, prohibiting the harvest of individuals of a certain size, and restricting hunting seasons. An interesting debate among administrators of different regional agencies (environmental and fisheries) has recently arisen for sea urchins because the demand and economic interest are evidently contrasting with the species conservation and harvest sustainability. Because of the possible cumulating effects of multi predators on sea urchins, this should only be achieved by considering a stage-structure and predator-urchin modeling (Panja, 2018) in order to provide context-dependent estimates of urchin stock assessment. This would also possibly lead to considering whether setting the threshold size for urchin harvest should depend on the overall sea urchin density and population structure (i.e., proportion of adults). Efforts addressing such goals should be encouraged to ensure sustainable resource exploitation on the basis of natural mortality, whose rate is expected to increase with reserve effects. Therefore, appropriate conservation measures are needed to contrast unsustainable harvesting, especially where other restrictions are been taken to achieve trophic upgrading. This would be in agreement with the general concern of stakeholders, since MPAs are known to achieve habitat conservation via halting fish harvest, rather than contributing to the further depletion of the stock (White et al., 2021) and changing the restricted harvest zones to no harvest has been widely discussed within the marine literature (Murray et al., 1999; Abecasis, Afonso & Erzini, 2015).

Conclusions

This study was aimed to evaluate the effect of human harvest of the herbivore sea urchin P. lividus in protected areas, where its density is potentially already controlled by a higher natural predation risk, by comparing effects of harvest outside protected areas, where predation risk is strongly reduced due to the low abundance of predatory fishes. By the spatial evaluation, sea urchin total and commercial densities in restricted harvest (RH, inside MPAs) sites were found less abundant than those at no harvest (NH, MPA reserves) and unrestricted (UH, outside MPAs) sites. These patterns are likely due to the cumulative effect of human harvest and natural predation on sea urchin density, even if the lack of quantitative data about predation rate prevent any conclusive statement. The assessment of predation risk through adequate sea urchin natural predatory experiments may also represent the goal of further investigations. However, results suggest that complex mechanisms, including synergistic effects between natural biotic interactions and human pressures, may occur on sea urchin populations, and the assessment of MPA effects on P. lividus populations would be crucial to guide management decisions on regulating harvest permits.

Furthermore, by the temporal evaluation, it was estimated the rate of change in sea urchin density over time indicating that only NH conditions promoted the increase of sea urchin density and that current local management of the MPAs has driven towards a regression of populations. Thus, enacting further rules to avoid extreme reductions in the MPAs, including banning or reducing sea urchin harvest and creating turnover of harvest zones depending on the local context, is encouraged. Also, effective assessment of sea urchin harvest landings to obtain accurate and reliable data on catches, enforced regulations and penalties, as well as local education workshops, may be useful tools to consider for achieving better conservation and sustainable targets.

Supplemental Information

Supplemental Information 1 Location of sampling sites in the Western Sardinia

Click here for additional data file.

Supplemental Information 2 Location of sampling sites in the North-Western Sardinia

Click here for additional data file.

Supplemental Information 3 Location of sampling sites in the Northern Sardinia

Click here for additional data file.

Supplemental Information 4 Location of sampling sites in the North-Eastern Sardinia

Click here for additional data file.

Supplemental Information 5 Location of sampling sites in the Southern Sardinia

Click here for additional data file.

Supplemental Information 6 The spatial evaluation and the temporal evaluation data

The total and commercial sea urchin abundance and the old and 2019 total abundance of sea urchins.

Click here for additional data file.

Supplemental Information 7 Raw data for the temporal evaluation

(1) restriction type; (2) site name; (3) old sea urchin abundance; (4) 2019 sea urchin abundance

Click here for additional data file.

We wish to thank Ms Katie Duff for the English revision of the manuscript.

Additional Information and Declarations

Competing Interests

Author Contributions

Data Availability

The authors declare there are no competing interests.

Giulia Ceccherelli conceived and designed the experiments, performed the experiments, prepared figures and/or tables, authored or reviewed drafts of the paper, and approved the final draft.

Piero Addis, Fabrizio Atzori, Nicoletta Cadoni, Marco Casu, Mario De Luca, G Andrea de Lucia, Nicola Fois, Francesca Frau, Vittorio Gazale Daniele Grech, Mariano Mariani, Massimo Salvatore Giovanni Marras, Augusto Giuseppe Navone, Arianna Pansini, Pieraugusto Panzalis, Federico Pinna, Alberto Ruiu and Fabio Scarpa performed the experiments, prepared figures and/or tables, and approved the final draft.

Stefania Coppa, Simone Farina and Ivan Guala performed the experiments, prepared figures and/or tables, authored or reviewed drafts of the paper, and approved the final draft.

Luigi Piazzi conceived and designed the experiments, performed the experiments, analyzed the data, prepared figures and/or tables, authored or reviewed drafts of the paper, and approved the final draft.

The following information was supplied regarding data availability:

The raw data are available in the Supplemental Files.

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
