# Peer review of "Sea urchin harvest inside marine protected areas: an opportunity to investigate the effects of exploitation where trophic upgrading is achieved"

_PeerJ, doi:10.7717/peerj.12971_

## Round 0.1 · original submission · Major Revisions

Two of the reviewers think the topic is interesting to include in PeerJ, but it has errors that need to be corrected before re-reviewing. The main issues to consider are:

(1) the potential for pseudoreplication to have influenced inferences,

(2) the need for more clarity and context regarding the motivation of the research,

(3) the need for additional information on the effect of the focal MPAs on predator populations, and

(4) an inability to replicate analyses using the provided data. Scientific writing needs to be improved and also to explain the main concerns of the Reviewers in methods and the experimental design. I think this will really improve the paper.

·

Basic reporting

The manuscript addresses an important topic, how cumulative effects of harvest and predator-driven impacts (resulting from harvest restrictions inside MPAs) can facilitate the control of sea urchin grazers. The manuscript is nicely written and draws on relevant studies. Nonetheless, I would suggest that the authors explicitly address, wherever relevant, all possible alternative explanations for the observed differences in total counts of urchins, such as differences in refuge availability between sites (i.e., sites with greater refuge space may hold more urchins), potential variation in recruitment success among sites, exposure (i.e., larger urchins are more susceptible to dislodgement at wave-exposed sites), etc. This could take the form of either an added sub-section within the Discussion, or brief explanations throughout the current Discussion text. Additionally, many of the implications of the study related to increased predator abundance inside MPAs are correlative and could be improved by providing specific examples where possible. While this study does not analyze predator diversity or abundance inside and outside of MPAs, the authors cite a few examples of other studies that do. The overall rationale for the cumulative effects of predators and harvest as drivers of urchin abundance could be improved by explicitly providing examples of known urchin predators that have increased inside of MPAs. I think this could be done relatively briefly without compromising the most parsimonious explanation of the patterns and processes driving observed changes in urchin abundance.

Experimental design

General comments on data collection: it would be helpful to know the general landscape rugosity within and across sites because if some sites are more rugose than others then those sites may be oversampled. For example, if the NH sites were more rugose and had more refuge space per quadrat sampled than the UH sites, then more available reef space would be sampled per quadrat within the NH sites. It would be nice to see whether substratum rugosity was consistent across all sampled sites. Moreover, MPAs are often designated around areas where habitat complexity (substrate type, rugosity, etc) is of interest, and this could potentially be a confounding effects if the non-MPA sites had a vastly different substrate type or complexity. The authors should specifically describe how sea urchins were quantified within each quadrat (e.g., by quantifying urchins from an ‘aerial’ view, or by searching in cracks and crevices within quadrats).

Lines 197-203 – the description of the sampling design could be improved by describing the spatial orientation of quadrat placement. For example, line 199 states that quadrats were “randomly placed meters of distance apart”. Were these quadrats haphazardly placed, or was selection criteria used to determine the quadrat placement?

Lines 198 – what is a sampling “site?” A figure could be helpful here to depict how quadrats were arranged and what other information was gathered at a single sampling site.

Lines 199 – The descriptions of the visual sampling methods could be improved by describing the criteria used for quantifying individuals. In other words, were all urchins of all sizes quantified? What about small (<2cm) and inconspicuous or cryptic urchins hidden in crevices?

Line 205 – “Methodology” is the study of methods. I recommend replacing “methodology” with “…consistency in the sampling methods used”

Line 206 – replace “75 could be” with “75 were ultimately…”

Line 15 – what is the rationale for using Euclidean distance?

Lines 216-220 – a brief table depicting the PERMANOVA design could be helpful.

Validity of the findings

The findings appear to fit within the scope of the study.

Additional comments

Line 253 – the first sentence of this section should be very impactful, based on the results. I would suggest editing this sentence to reflect the major take-home implications of this study.

Line 254-258 – these sentences are restating what is already in the results. I suggest the authors move the following sentence (line 259) to an earlier place in the paragraph because this is a major implication of the study.

Line 267 – “proved” implies absolute certainty. Given the observational design (rather than experimental) of this study, I suggest the authors replace “proved” with “evidenced by…” This is one place where the authors might consider incorporating alternative hypotheses.

Line 285 – what about behavioral responses? Is it possible that the pronounced increase at NH sites could be due to a behavioral shift, where urchins emerged from crypsis and became more detectible? Several physical and biotic factors are known to cue these types of behavioral responses.

Line 288 – the statement “harvest restriction remains the most relevant driver affecting sea urchins” requires a citation.

Lines 313-327 – this final paragraph introduces several important conservation and management concerns, but it diminishes the overall impact of the study. I recommend that the authors conclude the manuscript by revising the final paragraph to a short succinct statement that focuses on the one or two major implications resulting from this study.

Figure 2 – confidence intervals would be useful to visually identify significant differences.

Figure 4 – this is a really important figure. I suggest the authors depict the temporal trends as a line chart with a separate line for each harvest type, rather than a histogram. Additionally, the dramatic increase in total abundance at the NH site in 2019 is interesting. Why the increase in this year only?

Figure 5 – how the rate of variation was calculated is not entirely clear. Is this cumulative across all years? If so, is it possible that the 2019 increase at the NH sites could be driving the observed overall net increase in rate of variation?

Reviewer 2 ·

Basic reporting

This manuscript describes analyses performed on a series of long-term MPA data sets collected by the authors to assess whether a decrease in urchin populations has occurred within restricted harvest MPA zones. The authors make the case that urchins in restricted harvest zones are subject to double the predation pressure from both natural predators and human harvest; natural predators affecting urchins up to ~40 mm and humans harvesting those greater than ~50 mm. Indeed, this is cause for concern as MPAs are well-known for their ability to prevent kelp forest loss via halting harvest (White et al. 2020) rather than contributing to the further depletion of the stock. Consistent with that argument the authors go on to make the recommendation of changing the restricted harvest zones to no harvest, which is a large topic of conversation within the marine conversation literature (Abecasis et al. 2015; Murray et al. 1999). I do believe the study at its core can offer key management guidance; however, I believe some areas need to be addressed to make these contributions clearer. Below I have provided my comments and suggestions, in the appropriate sections, in hopes that they may help strengthen the manuscript.

Major Comments

1. Clarity on the motivation of the research

While reading about the motivation of the study and throughout the discussion, I was confused as to why the underabundance of urchins is a concern for the focal area(s). Much of the literature on kelp forests is typically concerned with the overabundance of urchins in kelp forest ecosystems and what their impact on the ecosystem will be (Filbee-Dexter and Scheibling, 2014). I am familiar with the concern of urchin underabundance in coral reef ecosystems and how their lack of presence can lead to an overgrowth of algae in the coral dominated ecosystems (Nozawa et al. 2020), but was not entirely sure if that was the case here, aside from the brief mention on line 84. I believe the manuscript would greatly benefit from this background information and context being more explicitly stated.

2. Additional information on MPA success

I am concerned by the lack of data (or cited information) on the success of the focal MPA zones in reviving the local predator population. Especially in the areas of restricted take, are key urchin predators also being fished? If so, would that then not make the urchin populations in restricted take areas only affected by human hunting, rather than the dual effect of both human and natural predator(s)? Given that the authors’ strong recommendation of changing restricted harvest sites to no take sites, it seems necessary to demonstrate that urchin predators have indeed increased in population within the restricted sites in the way that the manuscript presumes. I believe this could be demonstrated by either citing work that has been done in the focal MPAs (it appears Guidetti’s work may be appropriate to use if it includes the same MPAs), or by presenting data on the predator populations of the focal MPAs.



Minor comments

1. Presentation of data

I found it unnecessarily difficult to comprehend Figure 4. I think readers may benefit if the information was presented as a continuous time series plot with data presented as points with error bars rather than a bar graph.

2. Grammatical and spelling errors

There were grammatical and spelling errors throughout the manuscript. Some examples are:
• Filbee-Dexter and Scheibling, 2014 is misspelled both for the in-text and full citation
• Line 46: I believe “fasted” should be changed to “fastest”
• Line 221 – I believe “PARMANOVA” should be “PERMANOVA

3. Fleshed out figure captions

The figure captions could use additional detail explaining what data and analyses were used to generate the figure.

4. Additional citations

The manuscript would be strengthened if additional citations were added to some lines in the introduction and discussion sections. Some examples are:
• Line 69
• Line 315

Cited papers from all sections in review

Abecasis, D., Afonso, P. and Erzini, K., 2015. Toward adaptive management of coastal MPAs: The influence of different conservation targets and costs on the design of no-take areas. Ecological Informatics, 30, pp.263-270.

Filbee-Dexter, K. and Scheibling, R.E., 2014. Sea urchin barrens as alternative stable states of collapsed kelp ecosystems. Marine ecology progress series, 495, pp.1-25.

Hurlbert, S.H., 1984. Pseudoreplication and the design of ecological field experiments. Ecological monographs, 54(2), pp.187-211.

Murray, S.N., Ambrose, R.F., Bohnsack, J.A., Botsford, L.W., Carr, M.H., Davis, G.E., Dayton, P.K., Gotshall, D., Gunderson, D.R., Hixon, M.A. and Lubchenco, J., 1999. No-take reserve networks: sustaining fishery populations and marine ecosystems. Fisheries, 24(11), pp.11-25.

Nozawa, Y., Lin, C.H. and Meng, P.J., 2020. Sea urchins (diadematids) promote coral recovery via recruitment on Taiwanese reefs. Coral Reefs, 39, pp.1199-1207.

White, J.W., Yamane, M.T., Nickols, K.J. and Caselle, J.E., 2021. Analysis of fish population size distributions confirms cessation of fishing in marine protected areas. Conservation Letters, 14(2), p.e12775.

Experimental design

Major Comments

1. Potential pseudoreplication and clarity on statistical interpretation

I am concerned by the potential consequence of not having accounted for pseudoreplication (Hurlbert, 1984). In the analyses performed, the statistics were run on the site level, however inference is written on MPA zone type. This raises concern as there are multiple sites in one zone type within the same MPA which appear to have been incorrectly interpreted as individual statistical replicates. My suggestions are to either re-run the analyses on the MPA zone-level, where each zone within each MPA is a replicate (by having averaged quadrats and sites within each MPA or zone), or perform an appropriate hierarchical analysis (e.g., nested ANOVA if residuals are unbiased and homoscedastic) to permit inferences at the level of MPA zone.

2. Repeatable analyses

I was confused on the naming convention of the data provided in the supplemental material. The data appears to be a version that might be ready to plug into PERMANOVA or another statistical analysis, but there are no formulas or complete data for the various sites that were monitored. The data are not “raw” (quadrat-level) data, as is implied by the filename.

Validity of the findings

Due to the potential consequence of not having accounted for pseudoreplication (see above section) I am concerned for how that statistical conclusion may have influenced the inference.

·

Basic reporting

This article should be reviewed by a person who is a native English speaker as English is very poor and the terminology used is ambiguous and not very technical.

Scientific writing needs to be seriously improved.

Authors confuse density and abundance in the manuscript, figures, and tables.

There is no information on natural predation to compare with harvesting.

Experimental design

The sampling methodology is vague.

Validity of the findings

The discussion is weak.

Additional comments

Specific comments have been included in the PDF

---

## Round 0.2 · Minor Revisions

The authors now incorporate a biomass estimate of two predatory fish species inside and outside of harvested areas, but these biomass estimates are not enough support for their main conclusion of cumulative effects between natural predation and harvest. It is necessary to show data on their interaction strengths, urchin consumption rates, or cite literature that provides this information. Please rewrite your conclusion according to both reviewers' suggestions.

·

Basic reporting

no comment

Experimental design

no comment

Validity of the findings

no comment

Additional comments

no comment

·

Basic reporting

The manuscript has been improved but several points should be taken into account (attached pdf file).

Since you don’t have data on natural predation, I would not stress that in the discussion. I will recommend using it at the end as a potential factor influencing the sea urchins densities due to the larger number of predatory fish in MPAs compared with unprotected areas.

Why do fishermen prefer harvesting in RH compared with UH sites?

Experimental design

There are significant differences in the spatial-temporal variability but no significant differences among years. A statistical test will provide robust discussion and conclusions.

PRIMER V6 is mentioned in the references but not cited in the manuscript. PERMANOVA is an extension of PRIMER and was used in the analyses but is not mentioned in the references.

Validity of the findings

"no comment"

---

## Round 0.3 · accepted · Accept

I believe that the authors have made the main changes suggested by the reviewers and justified those that they decided not to make. I am satisfied with the changes made so the work is ready for publication